# TADAM: Task dependent adaptive metric for improved few-shot learning

**Boris N. Oreshkin**
Element AI
boris@elementai.com

**Pau Rodriguez**
Element AI, CVC-UAB
pau.rodriguez@elementai.com

**Alexandre Lacoste**
Element AI
allac@elementai.com

## Abstract

Few-shot learning has become essential for producing models that generalize from few examples. In this work, we identify that metric scaling and metric task conditioning are important to improve the performance of few-shot algorithms. Our analysis reveals that simple metric scaling completely changes the nature of few-shot algorithm parameter updates. Metric scaling provides improvements up to 14% in accuracy for certain metrics on the mini-Imagenet 5-way 5-shot classification task. We further propose a simple and effective way of conditioning a learner on the task sample set, resulting in learning a task-dependent metric space. Moreover, we propose and empirically test a practical end-to-end optimization procedure based on auxiliary task co-training to learn a task-dependent metric space. The resulting few-shot learning model based on the task-dependent scaled metric achieves state of the art on mini-Imagenet. We confirm these results on another few-shot dataset that we introduce in this paper based on CIFAR100.

## 1 Introduction

Humans can learn to identify new categories from few examples, even from a single one [2]. Few-shot learning has recently attracted significant attention [33, 28, 29, 24, 17, 16], as it aims to produce models that can generalize from small amounts of labeled data. In the few-shot setting, one aims to learn a model that extracts information from a set of support examples (*sample* set) to predict the labels of instances from a *query* set. Recently, this problem has been reframed into the meta-learning framework [22], *i.e.* the model is trained so that given a *sample* set or task, produces a classifier for that specific task. Thus, the model is exposed to different tasks (or episodes) during the training phase, and it is evaluated on a non-overlapping set of new tasks [33].

Two recent approaches have attracted significant attention in the few-shot learning domain: *Matching Networks* [33], and *Prototypical Networks* [28]. In both approaches, the *sample* set and the *query* set are embedded with a neural network, and nearest neighbor classification is used given a metric in the embedded space. Since then, the problem of learning the most suitable metric for few-shot learning has been of interest to the field [33, 28, 29, 17, 16]. Learning a metric space in the context of few-shot learning generally implies identifying a suitable similarity measure (e.g. cosine or Euclidean), a feature extractor mapping raw inputs onto similarity space (*e.g.* convolutional stack for images or LSTM stack for text), a cost function to drive the parameter updates, and a training scheme (often episodic). Although the individual components in this list have been explored, the relationships between them have not received considerable attention.

In the current work we aim to close this gap. We show that taking into account the interaction between the identified components leads to significant improvements in the few-shot generalization. In particular, we show that a non-trivial interaction between the similarity metric and the cost function can be exploited to improve the performance of a given similarity metric via scaling. Using this mechanism we close more than the 10% gap in performance between the cosine similarity and

the Euclidean distance reported in [28]. Even more importantly, we extend the very notion of the metric space by making it task dependent via conditioning the feature extractor on the specific task. However, learning such a space is in general more challenging than learning a static one. Hence, we find a solution in exploiting the interaction between the conditioned feature extractor and the training procedure based on auxiliary co-training on a simpler task. Our proposed few-shot learning architecture based on task-dependent scaled metric achieves superior performance on two challenging few-shot image classification datasets. It shows up to 8.5% absolute accuracy improvement over the baseline (Snell et al. [28]), and 4.8% over the state-of-the-art [17] on the 5-shot, 5-way mini-Imagenet classification task, reaching 76.7% of accuracy, which is the best-reported accuracy on this dataset.

## 1.1 Background

We consider the episodic $M$-shot, $K$-way classification scenario. In this scenario, a learning algorithm is provided with a *sample* set $\mathcal{S} = \{(\mathbf{x}_i, y_i)\}_{i=1}^{MK}$ consisting of $M$ examples for each of $K$ classes and a *query* set $\mathcal{Q} = \{(\mathbf{x}_i, y_i)\}_{i=1}^{q}$ for a task to be solved within a given episode. The *sample* set provides the task information via observations $\mathbf{x}_i \in \mathbb{R}^{D_\mathbf{x}}$ and their respective class labels $y_i \in \{1, \ldots, K\}$. Given the information in the *sample* set $\mathcal{S}$, the learning algorithm is able to classify individual samples from the *query* set $\mathcal{Q}$. Next, we define a similarity measure $d : \mathbb{R}^{D_\mathbf{z} \times D_\mathbf{z}} \to \mathbb{R}$. Note that $d$ does not have to satisfy the classical metric properties (non-negativity, symmetry, subadditivity) to be useful in the context of few-shot learning. The dimensionality of metric input, $D_\mathbf{z}$, will most naturally be related to the size of embedding created by a (deep) feature extractor $f_\phi : \mathbb{R}^{D_\mathbf{x}} \to \mathbb{R}^{D_\mathbf{z}}$, parameterized by $\phi$, mapping $\mathbf{x}$ to $\mathbf{z}$. Here $\phi \in \mathbb{R}^{D_\phi}$ is a list of parameters defining $f_\phi$, *e.g.* a list of weights in a neural network. The set of representations $(f_\phi(\mathbf{x}_i), y_i), \forall (\mathbf{x}_i, y_i) \in \mathcal{S}$ can directly be used to solve the few-shot learning classification problem by association. For example, Matching networks [33] use sample-wise attention mechanism to perform kernel label regression. Instead, Snell et al. [28] defined a feature representation $\mathbf{c}_k$ for each class $k$ as the mean over embeddings belonging to $\mathcal{S}_k$: $\mathbf{c}_k = \frac{1}{K} \sum_{\mathbf{x}_i \in \mathcal{S}_k} f_\phi(\mathbf{x}_i)$. To learn $\phi$, they minimize $-\log p_\phi(y = k|\mathbf{x})$ using the softmax over prototypes $\mathbf{c}_k$ to define the likelihood: $p_\phi(y = k|\mathbf{x}) = \mathrm{softmax}(-d(f_\phi(\mathbf{x}), \mathbf{c}_k))$.

## 1.2 Summary of contributions

**Metric Scaling:** To our knowledge, this is the first study to (i) propose metric scaling to improve performance of few-shot algorithms, (ii) mathematically analyze its effects on objective function updates and (iii) empirically demonstrate its positive effects on few-shot performance.

**Task Conditioning:** We use a task encoding network to extract a task representation based on the task's *sample* set. This is used to influence the behavior of the feature extractor through FILM [19].

**Auxiliary task co-training:** We show that co-training the feature extraction on a conventional supervised classification task reduces training complexity and provides better generalization.

## 1.3 Related work

Three main approaches for solving the few-shot classification problem can be identified in the literature. The first one, which is used in this work, is the meta-learning approach, *i.e.* learning a model that, given a task (set of labeled data), produces a classifier that generalizes across all tasks [31, 25]. This is the case of Matching Networks [33], which optionally use a Recurrent Neural Network (RNN) to accumulate information about a given task. In MAML [6], the parameters of an arbitrary learner model are optimized so that they can be quickly adapted to a particular task. In "Optimization as a model" [22], a learner model is adapted to a new episodic task by a recurrent meta-learner producing efficient parameter updates. A more general approach was proposed by Santoro et al. [24], where the meta-learner is trained to represent entries from a *sample* set in an external memory. Similarly, adaResNet [17] uses memory and the *sample* set to produce shift coefficients on the neuron activations of the *query* set classifier. Many recent approaches focus on learning a metric on the episodic feature space. Prototypical networks [28] use a feed-forward neural network to embed the task examples and perform nearest neighbor classification with the class centroids. The relation network approach by Sung et al. [29] introduces a separate learnable similarity metric. SNAIL [16] uses an explicit attention mechanism applicable both to supervised and to the sequence based reinforcement learning tasks. It has also been shown that these approaches benefit from leveraging unlabeled and simulated data [23, 34].

A second approach aims to maximize the distance between examples from different classes [10]. Similarly, in [7], a contrastive loss function is used to learn to project data onto a manifold that is invariant to deformations in the input space. In the same vein, in [5, 26, 30], triplet loss is used for learning a representation for few-shot learning. The attentive recurrent comparators [27] go beyond classical siamese approaches and use a recurrent architecture to learn to perform pairwise comparisons and predict if the compared examples belong to the same class.

The third approach relies on Bayesian modeling of the prior distribution of the different categories like in Li et al. [15], Bauer et al. [1], or Lake et al. [13], Edwards and Storkey [4], Lacoste et al. [12] who rely on hierarchical Bayesian modeling.

As for task conditioning, [3, 18, 19] proposed conditional batch normalization for style transfer and visual reasoning. Differently, we modify the conditioning scheme to adapt it to few-shot learning, introducing $\gamma_0, \beta_0$ priors, and auxiliary co-training. In the few-shot learning context, task conditioning ideas can be traced back to [33], although in an implicit form as there is no notion of task embedding. In our work, we explicitly introduce a task representation (see Fig. 1) computed as the mean of the task class centroids (task prototypes). This is much simpler than individual sample level LSTM/attention models in [33]. Conditioning in [33] is applied as a postprocessing of the output of a fixed feature extractor. We propose to condition the feature extractor by predicting its own batch normalization parameters thus making feature extractor behaviour task-dynamic without cumbersome fine-tuning on support set. In order to train the task conditioned architecture we use multitask training with a usual 64-way classification task. Even though auxiliary co-training is beneficial for learning in general, "little is known on *when* multitask learning works and whether there are data characteristics that help to determine its success" [20]. We show that combining task conditioning and auxiliary co-training is beneficial in the context of few-shot learning.

The scaling and temperature adjustment in the softmax was discussed by Hinton et al. [9] in the context of model distillation. We propose to use it in the context of the few-shot learning scenario and provide novel theoretical and empirical results quantifying the effects of scaling parameter.

The rest of the paper is organized as follows. Section 2 describes our contributions in detail. Section 3 highlights the importance of each contribution via an ablation study. The study is performed over two different benchmarks in the regime of 1-shot, 5-shot and 10-shot learning to verify if conclusions hold across different setups. Finally, Section 4 concludes the paper and outlines future research directions.

## 2 Model Description

### 2.1 Metric Scaling

Snell et al. [28] using approach described in detail in Section 1.1 found that the Euclidean distance outperformed the cosine distance used in Vinyals et al. [33]. We hypothesize that the improvement could be directly attributed to the interaction of the different scaling of the metrics with the softmax. Moreover, the dimensionality of the output is known to have a direct impact on the output scale even for the Euclidean distance [32]. Hence, we propose to scale the distance metric by a learnable temperature, $\alpha$, $p_{\phi,\alpha}(y = k|\mathbf{x}) = \text{softmax}(-\alpha d(\mathbf{z}, \mathbf{c}_k))$, to enable the model to learn the best regime for each similarity metric, thus improving the performances of all metrics. To further understand the role of $\alpha$, we analyze the class-wise cross-entropy loss function, $J_k(\phi, \alpha)$,[1]

$$J_k(\phi, \alpha) = \sum_{\mathbf{x}_i \in \mathcal{Q}_k} \Big[ \alpha d(f_\phi(\mathbf{x}_i), \mathbf{c}_k) + \log \sum_j \exp(-\alpha d(f_\phi(\mathbf{x}_i), \mathbf{c}_j)) \Big], \tag{1}$$

where $\mathcal{Q}_k = \{(\mathbf{x}_i, y_i) \in \mathcal{Q} : y_i = k\}$ is the *query* set corresponding to the class $k$. Its gradient, which is used to update parameters $\phi$ is given by the following expression:

$$\frac{\partial}{\partial \phi} J_k(\phi, \alpha) = \alpha \sum_{\mathbf{x}_i \in \mathcal{Q}_k} \left[ \frac{\partial}{\partial \phi} d(f_\phi(\mathbf{x}_i), \mathbf{c}_k) - \frac{\sum_j \exp(-\alpha d(f_\phi(\mathbf{x}_i), \mathbf{c}_j)) \frac{\partial}{\partial \phi} d(f_\phi(\mathbf{x}_i), \mathbf{c}_j)}{\sum_j \exp(-\alpha d(f_\phi(\mathbf{x}_i), \mathbf{c}_j))} \right]. \tag{2}$$

At first glance, the effect of $\alpha$ on the expression of the derivative is twofold: (i) an overall scaling, and (ii) regulating the sharpness of weighting in the second term inside the brackets on the RHS. Below we explore the behavior of the $\alpha$-normalized[2] gradient in the limits $\alpha \to 0$ and $\alpha \to \infty$.

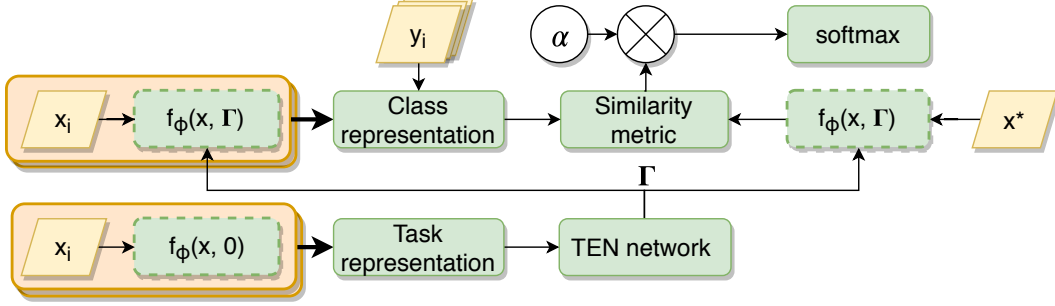

Figure 1: Proposed few-shot architecture. Blocks with shared parameters have dashed border.

**Lemma 1** (Metric scaling). *If the following assumptions hold:*

$$\mathcal{A}_1 : d(f_\phi(\mathbf{x}), \mathbf{c}_k) \neq d(f_\phi(\mathbf{x}'), \mathbf{c}_k), \forall k, \mathbf{x} \neq \mathbf{x}' \in \mathcal{Q}_k; \quad \mathcal{A}_2 : \left| \frac{\partial}{\partial \phi} d(f_\phi(\mathbf{x}), \mathbf{c}) \right| < \infty, \forall \mathbf{x}, \mathbf{c}, \phi,$$

*then it is true that:*

$$\lim_{\alpha \to 0} \frac{1}{\alpha} \frac{\partial}{\partial \phi} J_k(\phi, \alpha) = \sum_{\mathbf{x}_i \in \mathcal{Q}_k} \left[ \frac{K-1}{K} \frac{\partial}{\partial \phi} d(f_\phi(\mathbf{x}_i), \mathbf{c}_k) - \frac{1}{K} \sum_{j \neq k} \frac{\partial}{\partial \phi} d(f_\phi(\mathbf{x}_i), \mathbf{c}_j) \right], \quad (3)$$

$$\lim_{\alpha \to \infty} \frac{1}{\alpha} \frac{\partial}{\partial \phi} J_k(\phi, \alpha) = \sum_{\mathbf{x}_i \in \mathcal{Q}_k} \left[ \frac{\partial}{\partial \phi} d(f_\phi(\mathbf{x}_i), \mathbf{c}_k) - \frac{\partial}{\partial \phi} d(f_\phi(\mathbf{x}_i), \mathbf{c}_{j_i^*}) \right]; \quad (4)$$

*where $j_i^* = \arg \min_j d(f_\phi(\mathbf{x}_i), \mathbf{c}_j)$.*

*Proof.* Please refer to Appendix A. □

From Eq. (3), it is clear that for small $\alpha$ values, the first term minimizes the embedding distance between query samples and their corresponding prototypes. The second term maximizes the embedding distance between the samples and the prototypes of the non-belonging categories. For large $\alpha$ values (Eq. (4)), the first term is the same as in Eq. (3); while the second term maximizes the distance of the sample with the closest wrongly assigned prototype $\mathbf{c}_{j_i^*}$ (if any). If $j_i^* = k$ (no error), the derivative contribution of the point $\mathbf{x}_i$ is zero. This is equivalent to learning *only* from the hardest examples resulting in association errors. Thus, the two different regimes of $\alpha$ favor either minimizing the overlap of the sample distributions or correcting cluster assignments sample-wise.

The large $\alpha$ regime is more directly related to resolving the few-shot classification errors. At the same time, the update strategy generated in this regime has a drawback. As the optimization proceeds and the classification accuracy increases, the number of incorrectly classified samples reduces on average, and this leads to the reduction in the average effective batch size (more samples generate zero derivatives). Therefore, our hypothesis is that there is an optimal value of scaling parameter $\alpha$ for a given combination of dataset, metric and task. Section 3.4 empirically demonstrates that the optimal value of $\alpha$ indeed exists and it can be *e.g.* cross-validated on a validation set.

## 2.2 Task conditioning

Up until now we assumed the feature extractor $f_\phi(\cdot)$ to be task-independent. A dynamic task-conditioned feature extractor should be better suited for finding correct associations between given *sample* set class representations and query samples, this is implicitly done by Vinyals et al. [33] with a bidirectional LSTM as a postprocessing of a fixed feature extractor. Differently, we explicitly define a dynamic feature extractor $f_\phi(\mathbf{x}, \Gamma)$, where $\Gamma$ is the set of parameters predicted from a task representation such that the performance of $f_\phi(\mathbf{x}, \Gamma)$ is optimized given the task *sample* set $\mathcal{S}$. This is related to the FILM conditioning layer [19] and conditional batch normalization [3, 18] of the form $h_{\ell+1} = \boldsymbol{\gamma} \odot h_\ell + \beta$, where $\boldsymbol{\gamma}$ and $\boldsymbol{\beta}$ are scaling and shift vectors applied to the layer $h_\ell$. Concretely, we propose to use the mean of the class prototypes as the *task representation*, $\bar{\mathbf{c}} = \frac{1}{K} \sum_k \mathbf{c}_k$, encode it with a task embedding network (TEN), and predict layer-level element-wise scale and shift vectors $\boldsymbol{\gamma}, \boldsymbol{\beta}$ for each convolutional layer in the feature extractor (see Figures 1 and 2 in the Supplementary

Table 1: mini-Imagenet (Vinyals et al. [33]), 5-way classification results. [†]Our re-implementation.

|  | 1-shot | 5-shot | 10-shot |
|---|---|---|---|
| Meta Nets [22] | 43.4 | 60.6 | - |
| Matching Networks [33] | 46.6 | 60.0 | - |
| MAML [6] | 48.7 | 63.1 | - |
| Proto Nets [28] | 49.4 | 68.2 | $74.3^{\dagger}$ |
| Relation Net [29] | 50.4 | 65.3 | - |
| SNAIL [16] | 55.7 | 68.9 | - |
| Discriminative k-shot [1] | 56.3 | 73.9 | 78.5 |
| adaResNet [17] | 56.9 | 71.9 | - |
| Ours | **58.5** | **76.7** | **80.8** |

Materials, Section S1). The task representation defined as the mean of task class centroids (i) reduces the dimensionality of the TEN input and (ii) replaces expensive RNN/CNN/attention modeling. On the other hand, it is an effective way to cluster tasks. Tasks having larger number of similar classes in common will tend to cluster closer in the task representation space.

Our implementation of the TEN (see Supplementary Materials, Section S1 for more details) uses two separate fully connected residual networks to generate vectors $\gamma, \beta$. Following the terminology in [18], the $\gamma$ parameter is learned in the delta regime, *i.e.* predicting deviation from unity. The most critical component in being able to successfully train the TEN was the addition of the scalar $L_2$ penalized post-multipliers $\gamma_0$ and $\beta_0$. They limit the effect of $\gamma$ (and $\beta$) by encoding a prior belief that all components of $\gamma$ (and $\beta$) should be simultaneously close to zero for a given layer unless task conditioning provides a significant information gain for this layer. Mathematically, this can be expressed as $\beta = \beta_0 g_\theta(\bar{\mathbf{c}})$ and $\gamma = \gamma_0 h_\varphi(\bar{\mathbf{c}}) + 1$, where $g_\theta$ and $h_\varphi$ are predictors of $\beta$ and $\gamma$.

## 2.3 Architecture

The overall proposed few-shot classification architecture is depicted in Fig. 1 (see Supplementary Materials, Section S1 for more details). We employ ResNet-12 [8] as the backbone feature extractor. It has 4 blocks of depth 3 with 3x3 kernels and shortcut connections. 2x2 max-pool is applied at the end of each block. Convolutional layer depth starts with 64 filters and is doubled after every max-pool. Note that this architecture is similar in spirit to architectures used in [1] and [17], but we do not use any projection layers before or after the main backbone ResNet. On the first pass over sample set, the TEN predicts the values of $\gamma$ and $\beta$ parameters for each convolutional layer in the feature extractor from the task representation. Next, the *sample* set and the *query* set are processed by the feature extractor conditioned with the values of $\gamma$ and $\beta$ just generated. Both outputs are fed into a similarity metric to find an association between class prototypes and query instances. The output of similarity metric is scaled by scalar $\alpha$ and is fed into a softmax layer.

## 2.4 Auxiliary task co-training

The TEN (Section 2.2) introduces additional complexity into the architecture via task conditioning layers inserted after the convolutional and batch norm blocks. We empirically observed that simultaneously optimizing convolutional filters and the TEN is overly challenging. We solved the problem by auxiliary co-training with an additional logit head (the normal 64-way classification in mini-Imagenet case). The auxiliary task is sampled with a probability that is annealed over episodes. We annealed it using an exponential decay schedule of the form $0.9^{\lfloor 20t/T \rfloor}$, where $T$ is the total number of training episodes, $t$ is episode index. The initial auxiliary task selection probability was cross-validated to be 0.9 and the number of decay steps was chosen to be 20. We observed significant positive effects from the auxiliary task co-training (please refer to Section 3.4). The same positive effects were not observed with simple pre-training of the feature extractor. We attribute this to the regularization effects achieved via back-propagating auxiliary task gradients together with those of the main task.

It is of interest to note that the few-shot co-training with an auxiliary classification task is related to curriculum learning [24]. The auxiliary classification problem could be considered a part of a simpler curriculum that helps the learner acquire minimal skill level necessary before tackling on harder

few-shot classification tasks. Being effective at feature extraction (i.e. at task representation) forms a "prerequisite" at being effective at re-conditioning features based on the representation of a given task.

# 3   Experimental Results

Table 1 presents our key result in the context of existing state-of-the art. The five first rows show approaches that use the same feature extractor as [33], *i.e.* four stacked convolutions layers of 64 filters (32 in [22, 6] to avoid overfitting). In the following rows we include models like the one we propose, which is based on resnet [8]. Concretely, SNAIL [16], adaResNet [17], and our architecture use four residual blocks of three stacked $3 \times 3$ convolutional layers, each block followed by max pooling. Differently, the feature extractor proposed in [1] is based on a ResNet-34 architecture with a reduced number of features.

As it can be seen, the proposed algorithm significantly improves over the existing state-of-the-art results on the mini-Imagenet dataset. In the rest of the section we address the following research questions: (i) can metric scaling improve few-shot classification results? (Sections 3.2 and 3.4), (ii) what are the contributions of each components of our proposed architecture? (Section 3.4), (iii) can task conditioning improve few-shot classification results and how important it is at different feature extractor depths? (Sections 3.3 and 3.4), and (iv) can auxiliary classification task co-training improve accuracy on the few-shot classification task? (Section 3.4).

## 3.1   Experimental setup and datasets

The details of the experimental and training setup are provided in Supplementary Materials, Section S3. Note that we focused on mini-Imagenet [33] and Fewshot-CIFAR100 (introduced below) instead of Omniglot [14, 33, 28] as the former ones are more challenging, and the error rate is more sensitive to model improvements.

**mini-Imagenet.** The mini-Imagenet dataset was proposed by Vinyals et al. [33]. It has 100 classes, with 600 $84 \times 84$ images per class. Each task is generated by sampling 5 classes uniformly and 5 training samples per class, the remaining images from the 5 classes are used as query images to compute accuracy. To perform meta-validation and meta-test on unseen tasks (and classes), we isolate 16 and 20 classes from the original set of 100, leaving 64 classes for the training tasks. We use exactly the same train/validation/test split as the one suggested by Ravi and Larochelle [22].

**Fewshot-CIFAR100.** We introduce a new image based dataset based on CIFAR100 [11] for few-shot learning. We will refer to it as FC100. The main motivation for introducing this new dataset is to validate that the main results appearing in the experimental section generalize well beyond the mini-Imagenet. The secondary motivation is that the FC100 is suited for faster few-shot scenario prototyping than the mini-Imagenet and it presents a more challenging few-shot learning problem, because of reduced image size. On top of that, we propose a class split in FC100 to minimize the information overlap between splits to make it significantly more challenging than *e.g.* Omniglot. The original CIFAR100 dataset consists of $32 \times 32$ color images belonging to 100 different classes, 600 images per class. The 100 classes are further grouped into 20 superclasses. We split the dataset by superclass, rather than by individual class to minimize the information overlap. Thus the train split contains 60 classes belonging to 12 superclasses, the validation and test contain 20 classes belonging to 5 superclasses each. The exact class split is provided in Supplementary Materials, Section S2. The tasks are sampled uniformly at random within train, validation and test subsets. Therefore, each task with high probability contains samples belonging to classes from several superclasses.

## 3.2   On the similarity metric

We re-implemented prototypical networks [28], and use the Euclidean and the cosine similarity to test the effects of scaling (see Section 2). We closely follow the experimental setup defined by Snell et al. [28] (same feature extractor and training procedure). The scaling parameter $\alpha$ used on the last row was cross-validated on the validation set. Results are presented in Table 2.

Table 2: Average classification accuracy in percent with 95% confidence interval. 5-shot, 5 way classification task. The three last rows correspond to our implementation, first with euclidean distance, second with cosine distance, and third with the scaled cosine distance.

| | mini-Imagenet | | FC100 | |
| --- | --- | --- | --- | --- |
| | 5-way train | 20-way train | 5-way train | 20-way train |
| Proto Nets [28] | $65.8 \pm 0.7$ | $68.2 \pm 0.7$ | N/A | N/A |
| Proto Nets | $67.7 \pm 0.2$ | $68.9 \pm 0.3$ | $51.1 \pm 0.2$ | $50.3 \pm 0.3$ |
| Prototypical Cosine | $54.5 \pm 1.1$ | $53.9 \pm 0.6$ | $40.9 \pm 0.6$ | $37.1 \pm 1.9$ |
| Prototypical Cosine Scaled | $68.2 \pm 0.8$ | $68.1 \pm 0.7$ | $51.0 \pm 0.6$ | $49.6 \pm 0.5$ |

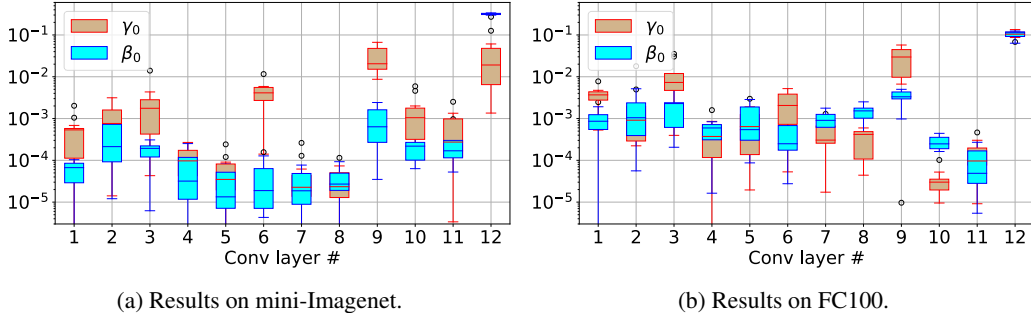

(a) Results on mini-Imagenet.  (b) Results on FC100.

Figure 2: Distribution of the absolute values of the TEN scaling and bias parameters $\gamma_0$ and $\beta_0$ across layers of ResNet feature extractor. X-axes depict layer number in both subplots. Higher convolutional layers are located closer to the final softmax layer.

As it can be seen in row two of Table 2, our re-implementation of Proto Nets [28] obtained slightly better performance (68.9% and 67.7%) in 20-way and 5-way training scenarios respectively by increasing the number of training steps from 20K to 40K[3].

Importantly, we confirm the hypothesis that the improvement attributed to the Euclidean distance in [28] was due to a scaling effect. Namely, we show that the scaled cosine similarity matches very closely the performance of the Euclidean metric, with an improvement of 14 percentage points on the mini-Imagenet (similar results on FC100) over the non-scaled version. In order to control for the potential effect that the scaling parameter $\alpha$ may have on the learning rate as indicated by Equation (2) training was performed using multiple initial learning rates (covering the range between 0.0005 and 0.01), obtaining similar accuracy each time. Hereinafter, we report the results with the Euclidean metric for brevity, since the cosine produces similar results. Moreover, since the prototypical approach with Euclidean distance as well as with the scaled cosine are close and both are superior to [33], we base our results on [28].

### 3.3 TEN importance across layers

We hypothesized in Section 2.2 that the TEN conditioning should not be equally important at all depths. Fig. 2 depicts the boxplot of the empirical observations of the learned TEN post-multipliers[4] $\gamma_0$ and $\beta_0$ at different depths of the feature extractor. We can see that for the multiplier $\gamma$, the absolute value of its scale $\gamma_0$ tends to increase as we approach the softmax layer. Interestingly, peaks can be observed every 3 layers (layers 3, 6, 9, 12). The peaks correspond to the location of the convolutional layers preceding the max-pool layers. For the bias parameter $\beta_0$, the only layer having a large absolute value of its scale is the last layer, before the softmax. We attribute the observed pattern to the fact that the shallower layers in the feature extractor tend to be less task-specific than the deeper layers. Following this intuition, we performed experiments in which we (i) kept the TEN injection solely in layers preceding the max pool and (ii) kept the TEN injection only in the very last layer. Interestingly,

Table 3: Average classification accuracy (%) with 95% confidence interval on the 5 way classification task, and training with the Euclidean distance. The scale parameter is cross-validated on the validation set. AT: auxiliary co-training. TC: task conditioning with TEN.

| $\alpha$ | AT | TC | mini-Imagenet | | | FC100 | | |
|---|---|---|---|---|---|---|---|---|
| | | | 1-shot | 5-shot | 10-shot | 1-shot | 5-shot | 10-shot |
| | | | $56.5 \pm 0.4$ | $74.2 \pm 0.2$ | $78.6 \pm 0.4$ | $37.8 \pm 0.4$ | $53.3 \pm 0.5$ | $58.7 \pm 0.4$ |
| ✓ | | | $56.8 \pm 0.3$ | $75.7 \pm 0.2$ | $79.6 \pm 0.4$ | $38.0 \pm 0.3$ | $54.0 \pm 0.5$ | $59.8 \pm 0.3$ |
| ✓ | ✓ | | $58.0 \pm 0.3$ | $75.6 \pm 0.4$ | $80.0 \pm 0.3$ | $39.0 \pm 0.4$ | $54.7 \pm 0.5$ | $60.4 \pm 0.4$ |
| ✓ | | ✓ | $54.4 \pm 0.3$ | $74.6 \pm 0.3$ | $78.7 \pm 0.4$ | $37.8 \pm 0.2$ | $54.0 \pm 0.7$ | $58.8 \pm 0.3$ |
| ✓ | ✓ | ✓ | $\mathbf{58.5 \pm 0.3}$ | $\mathbf{76.7 \pm 0.3}$ | $\mathbf{80.8 \pm 0.3}$ | $\mathbf{40.1 \pm 0.4}$ | $\mathbf{56.1 \pm 0.4}$ | $\mathbf{61.6 \pm 0.5}$ |

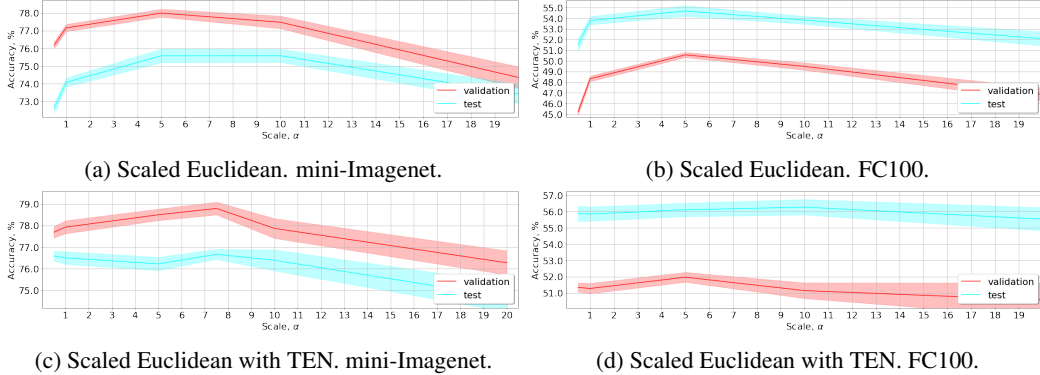

(a) Scaled Euclidean. mini-Imagenet.  (b) Scaled Euclidean. FC100.

(c) Scaled Euclidean with TEN. mini-Imagenet.  (d) Scaled Euclidean with TEN. FC100.

Figure 3: Metric scale parameter $\alpha$ cross-validation results.

we saw that TEN layers with small weight still provide some positive contribution, although most of the contribution is indeed provided by the layers preceding the max pool operation.

## 3.4 Ablation study

In this section, we study the impact in generalization accuracy of the scaling, task conditioning, auxiliary co-training, and the feature extractor. Results are summarized in Table 3.

First, we validated the hypothesis that there is an optimal value of the metric scaling parameter ($\alpha$) for a given combination of dataset and metric, which is reflected in the inverse U-shape of the curves in Fig. 3.

Second, we studied the effects of the task conditioning described in Section 2.2. No improvement was observed for the task-conditioned ResNet-12 without auxiliary co-training (see Table 3). We observed that learning useful features for the TEN and the main feature extractor at the same time is hard and gets stuck in local extrema. The problem is solved by co-training on the auxiliary task of predicting Imagenet labels using an additional fully-connected layer with softmax, see Section 2.4. In effect, we observed that auxiliary co-training provides two benefits: (i) making the initial convergence easier, and (ii) providing regularization on the few-shot learning task by forcing the feature extractor to perform well on two decoupled tasks. The latter benefit can only be observed when the feature extraction unit is sufficiently decoupled on the main task and the auxiliary task via the use of TEN (the feature extractor output is additionally adjusted on the target task using FILM).

As it can be seen in the last row of Tables 1 and 3, our model trained with TEN and auxiliary co-training outperforms all the baselines and achieves state-of-the-art results.

## 4 Conclusions and Future Work

We proposed, analyzed, and empirically validated several improvements in the domain of few-shot learning. We showed that the scaled cosine similarity performs at par with Euclidean distance, unlike its unscaled counterpart. In fact, based on our results, we argue that the scaling factor is a

necessary standard component of any few-shot learning algorithm relying on a similarity metric and the cross-entropy loss function. This is especially important in the context of finding new more effective similarity measures for few-shot learning. Moreover, our theoretical analysis demonstrated that simply scaling the similarity metric results in completely different regimes of parameter updates when using softmax and categorical cross-entropy. We also identified that the optimal performance is achieved in between two asymptotic regimes of the softmax. This poses the research question of explicitly designing loss functions and the $\alpha$ schedules optimal for few-shot learning. We further proposed task representation conditioning as a way to improve the performance of a feature extractor on the few-shot classification task. In this context, designing more powerful task representations, for example, based on higher order statistics of class embeddings, looks like a very promising venue for future work. The experimental results obtained on two independent challenging datasets demonstrated that the proposed approach significantly improves over existing results and achieves state-of-the-art on few-shot image classification task.

## Appendix

## A    Proof of Lemma 1

First, consider the case $\alpha \to 0$. Denoting $\mathbf{z}_i^\phi = f_\phi(\mathbf{x}_i)$ we have:

$$
\begin{aligned}
\lim_{\alpha \to 0} \frac{1}{\alpha} \frac{\partial}{\partial \phi} J_k(\phi, \alpha) &= \sum_{\mathbf{x}_i \in \mathcal{Q}_k} \frac{\partial}{\partial \phi} d(\mathbf{z}_i^\phi, \mathbf{c}_k) - \lim_{\alpha \to 0} \frac{\sum_j \exp(-\alpha d(\mathbf{z}_i^\phi, \mathbf{c}_j)) \frac{\partial}{\partial \phi} d(\mathbf{z}_i^\phi, \mathbf{c}_j)}{\sum_j \exp(-\alpha d(\mathbf{z}_i^\phi, \mathbf{c}_j))} \\
&= \sum_{\mathbf{x}_i \in \mathcal{Q}_k} \frac{\partial}{\partial \phi} d(\mathbf{z}_i^\phi, \mathbf{c}_k) - \frac{1}{K} \sum_j \frac{\partial}{\partial \phi} d(\mathbf{z}_i^\phi, \mathbf{c}_j) \\
&= \sum_{\mathbf{x}_i \in \mathcal{Q}_k} \frac{K-1}{K} \frac{\partial}{\partial \phi} d(\mathbf{z}_i^\phi, \mathbf{c}_k) - \frac{1}{K} \sum_{j \neq k} \frac{\partial}{\partial \phi} d(\mathbf{z}_i^\phi, \mathbf{c}_j).
\end{aligned}
$$

Second, consider the case $\alpha \to \infty$:

$$
\begin{aligned}
\lim_{\alpha \to \infty} \frac{1}{\alpha} \frac{\partial}{\partial \phi} J_k(\phi, \alpha) &= \sum_{\mathbf{x}_i \in \mathcal{Q}_k} \frac{\partial}{\partial \phi} d(\mathbf{z}_i^\phi, \mathbf{c}_k) - \sum_j \lim_{\alpha \to \infty} \frac{\exp(-\alpha d(\mathbf{z}_i^\phi, \mathbf{c}_j)) \frac{\partial}{\partial \phi} d(\mathbf{z}_i^\phi, \mathbf{c}_j)}{\sum_\ell \exp(-\alpha d(\mathbf{z}_i^\phi, \mathbf{c}_\ell))} \\
&= \sum_{\mathbf{x}_i \in \mathcal{Q}_k} \frac{\partial}{\partial \phi} d(\mathbf{z}_i^\phi, \mathbf{c}_k) - \sum_j \lim_{\alpha \to \infty} \frac{\frac{\partial}{\partial \phi} d(\mathbf{z}_i^\phi, \mathbf{c}_j)}{1 + \sum_{\ell \neq j} \exp(-\alpha [d(\mathbf{z}_i^\phi, \mathbf{c}_\ell) - d(\mathbf{z}_i^\phi, \mathbf{c}_j)])}.
\end{aligned}
$$

It is obvious that whenever at least one of the exponential terms in the denominator in the expression above has positive rate, corresponding to the case $\exists \ell \neq j : [d(\mathbf{z}_i^\phi, \mathbf{c}_\ell) - d(\mathbf{z}_i^\phi, \mathbf{c}_j)] < 0$, the ratio converges to zero as $\alpha \to \infty$ under assumption $\mathcal{A}_2$. The only case when the limit is non-zero is when $\mathbf{c}_j$ is the prototype closest to the query point $\mathbf{x}_i$. If we define the index of this prototype as $j_i^* = \arg\min_j d(\mathbf{z}_i^\phi, \mathbf{c}_j)$, then the following holds: $\forall \ell \neq j_i^* : [d(\mathbf{z}_i^\phi, \mathbf{c}_\ell) - d(\mathbf{z}_i^\phi, \mathbf{c}_{j_i^*})] > 0$, leading (under additional assumption $\mathcal{A}_1$) to:

$$
\lim_{\alpha \to \infty} \frac{1}{1 + \sum_{\ell \neq j} \exp(-\alpha [d(\mathbf{z}_i^\phi, \mathbf{c}_\ell) - d(\mathbf{z}_i^\phi, \mathbf{c}_{j_i^*})])} = 1.
$$

Therefore, (4) follows.                                                                 □

## Acknowledgements

Authors acknowledge the support of the Spanish project TIN2015-65464-R (MINECO/FEDER), the 2016FI B 01163 grant of Generalitat de Catalunya. Authors would like to thank Nicolas Chapados, Adam Salvail and Rachel Samson as well as anonymous reviewers for their careful reading of the manuscript and for providing constructive feedback and valuable suggestions.

## Footnotes

[1]Note that the total loss is simply $J(\phi, \alpha) = \sum_k J_k(\phi, \alpha)$

[2]The effect of $\alpha$-related gradient scaling is trivial.

[3]With 20K steps it was possible to recover the exact original performance reported in Snell et al. [28], which is not included in Table 2 for the sake of brevity.

[4]Larger absolute values of $\gamma_0$ and $\beta_0$ imply a larger influence of their respective TEN layers

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
