[Supplementary Material]

# Supplementary Materials. TADAM: Task dependent adaptive metric for improved few-shot learning

## S1  Architecture details

(a) Convolutional block with TEN.          (b) Resnet block with TEN.

Figure 1: Components of the ResNet-12 feature extractor.

**ResNet-12 architecture details.** The resnet blocks used in the ResNet-12 feature extractor are shown in Fig. 1. The feature extractor consists of 4 resnet blocks shown in Fig. 1b followed by a global average-pool. Each resnet block consists of 3 convolutional blocks shown in Fig. 1a followed by 2x2 max-pool. Each convolutional layer is followed by a batch norm layer and the swish-1 activation function proposed by Ramachandran et al. [21]. We found that the fully convolutional architecture performs best as a few-shot feature extractor, both on mini-Imagenet and on FC100. We found that inserting additional projection layers after the ResNet stack was always detrimental to the few-shot performance. We cross-validated this result with multiple hyper-parameter settings for the projection layers (number of layers, layer widths, and dropout). In addition to that, we observed that adding extra convolutional layers and max-pool layers before the ResNet stack was detrimental to the few-shot performance. Therefore, we used fully convolutional, fully residual architecture in all our experiments.

The hyperparameters for the convolutional layers are as follows. The number of filters for the first ResNet block was set to 64 and it was doubled after each max-pool block. The $L_2$ regularizer weight was cross-validated at 0.0005 for each layer.

**TEN architecture details.** The detailed architecture of the TEN block is depicted in Fig. 2. Our implementation of the TEN uses two separate fully connected residual networks to generate vectors $\gamma, \beta$. We cross-validated the number of layers to be 3. The first layer projects the task representation into the target width. The target width is equal to the number of filters of the convolutional layer that the TEN block is conditioning (see Fig. 1a). The remaining layers operate at the target width and each of them has a skip connection. The $L_2$ regularizer weight for $\gamma_0$ and $\beta_0$ was cross-validated at 0.01 for each layer. We found that smaller values led to considerable overfit. In addition to that, we were not able to successfully train TEN without $\gamma_0$ and $\beta_0$, because the training tended to be stuck in local minima where the overall effect of introducing TEN was detrimental to the few-shot performance of the architecture.

Figure 2: Architecture of the TEN block.

## S2    Few-shot CIFAR100 details

**Train split.** Super-class labels: {1, 2, 3, 4, 5, 6, 9, 10, 15, 17, 18, 19}; super-class names: {fish, flowers, food_containers, fruit_and_vegetables, household_electrical_devices, household_furniture, large_man-made_outdoor_things, large_natural_outdoor_scenes, reptiles, trees, vehicles_1, vehicles_2}.

**Validation split.** Super-class labels: {8, 11, 13, 16}; super-class names: {large_carnivores, large_omnivores_and_herbivores, non-insect_invertebrates, small_mammals}.

**Test split.** Super-class labels: {0, 7, 12, 14}; super-class names: {aquatic_mammals, insects, medium_mammals, people}.

We would like to stress that we still sample all the tasks uniformly at random within train, validation and test subsets. Therefore, each task with very high probability contains samples belonging to classes from several superclasses.

## S3    Training procedure details

**Episode composition.** The training procedure composes a few-shot training batch from several tasks, where a task is understood to be a fixed selection of 5 classes. We found empirically that for the 5-shot scenario the best number of tasks per batch was 2, for 10-shot it was 1 and for 1-shot it was 5. The *sample* set in each training batch was created using the same number of shots as in the target deployment (test) scenario. The images in the training *query* set were sampled uniformly at random. We observed that the best results were obtained when the number of query images was approximately equal to the total number of sample images in the batch. Thus we used 32 query images per task for 5-shot, 64 for 10-shot and 12 for 1-shot.

**The auxiliary classification task** is based on the usual 64-way training (for mini-Imagenet). Co-training uses a fixed batch of 64 image samples sampled uniformly at random from the training set. The learning rate annealing schedule for the auxiliary task is synchronized with that of the main few-shot task.

**Optimization, scheduling and learning rate.** When training with auxiliary classification task we used total 30000 episodes for training on mini-Imagenet and 10000 episodes for training on FC100. The results obtained with no auxiliary classification co-training used twice as many episodes. To obtain all our results we used SGD with momentum 0.9 and initial learning rate set at 0.1. The learning rate was annealed by a factor of 10 halfway through the training and two more times every 2500 episodes. The reported numbers are calculated using early-stopping based on validation set classification error tracking.

**Classification accuracy evaluation.** The accuracy is evaluated using 10 random restarts of the optimization procedure and based on 500 randomly generated tasks each having 100 random query samples.

**Reproducing results in [28].** To reproduce the results reported in [28] we used exactly the same setup and network architecture reported in the original paper.