[Reviews · NeurIPS 2018]

Reviewer 1



** After rebuttal Thank you for the author response. I have read the rebuttal and will maintain my score. The submission and rebuttal motivate three orthogonal improvements (metric scaling, task-conditioning, auxiliary task co-training) to Prototypical Networks. The paper would be much better written as three distinct units that evaluate the three components separately and in more depth. The mathematical analysis section of the paper consists of writing out the gradient of the objective function with the additional temperature parameter. Although this provides intuition **about the limiting cases of \alpha**, I would hesitate to call this a significantly "non-trivial" derivation. I also recommend that the authors provide empirical validation (e.g., plotting the contribution of \alpha-transformed gradients) for the non-limiting cases of \alpha in order to validate the intuition that varying \alpha in non-limiting cases trades off between "minimizing the overlap of the sample distributions or correcting cluster assignments sample-wise" (rather than just providing the final test performance as in Figure 3). I also do not agree that the generalization to a learned temperature parameter is exactly analogous to the development of Prototypical Networks from Matching Networks: While the Prototypical Network is itself a slight modification to the Matching Network, it touches on the important tradeoff between parametric and non-parametric density estimation (e.g., see the section on mixture density estimation in the Prototypical Networks paper). I do not see substantial evidence for an analogous claim that the form of the gradient with the additional learnable temperature parameter "has implications for future work in few-shot learning (FSL) and metric learning more generally." The submission needs more argumentation in order to convince the reader that the derivation would "drive algorithm design and empirical studies" more than the provided study that varies \alpha. Lastly, while it is nice to see evaluation on more datasets, the episodic training setup of CIFAR-100 is not sufficiently different from the few-shot miniImageNet dataset to be considered a major contribution to the community. ** Before the rebuttal This work proposes improvements to the prototypical network approach to few-shot classification introduced in [25]. The improvements are three-fold: (i) the addition of a learned temperature parameter in the computation of the softmax-normalized centroid distance; (ii) a modification to conditional normalization [16, 17] in which the conditioning is done as a function of the task-specific dataset as a whole; and (iii) an auxiliary prediction task implemented as an additional head in the architecture that outputs the logits for a 64-way classification task. The method gives better performance on the standardized miniImageNet benchmark as well as on a few-shot split of CIFAR100 introduced by the authors. QUALITY The theoretical contribution of this paper is to write out the gradient of the objective function from [25] with the addition of the temperature scaling parameter in the softmax computation. The limiting cases (zero and infinite temperature) are discussed for intuition even though they do not occur in practice as I assume the temperature parameter is transformed to be positive finite (although this is not stated). There are some flaws in quality throughout the paper: Several technical claims / hypotheses made in the paper are not evidenced: "Neural network initialization and batch norm encourages this regime." "If Df is large, the network may have to work outside of its optimal regime to be able to scale down the feature representation." "We hypothesize that the importance of TEN is not uniformly distributed over the layers in the feature extractor. The lower layers closer to the similarity metric need to be conditioned more than the shallower layers closer to the image because pixel-level features extracted by shallow level layers are not task specific." A conceptual point is made in the conclusion unnecessarily: it does not appear that the fact that "the scaled cosine similarity...does not belong to the class of Bregman divergences" is used anywhere in the paper, yet it is mentioned in the conclusion. CLARITY The paper is reasonably clear. Some points of are omitted: - Why not report results with temperature scaling and matching networks (instead of prototypical networks)? - What is the architecture of the TEN? What motivated this architecture (specifically, why use the centroids, which enable parameter sharing amongst the feature extractors, but still requires the TEN with additional parameters)? - What is the reason that the TEN "underperformed"? Was it overfitting? The experimental comparison in and the discussion of Table 1 does not identify the differences in architectural complexity between the reported methods (i.e., [1, 14, 15] and the proposed work employ a ResNet architecture while the other methods employ a shallow convolutional network). ORIGINALITY The architecture and algorithm are a combination of previously proposed methods (see "SIGNIFICANCE" below). The problem setup is not novel, although the authors apply the few-shot episode generation procedure of [30] to the CIFAR100 dataset. SIGNIFICANCE The work is incremental as a combination of previously proposed methods applied to prototypical networks for the task of few-shot classification. In particular: - "Metric scaling" is the addition of a learnable temperature parameter to the normalized distance computation in the regime of Matching Networks [30] or Prototypical Networks [25]. - "Task conditioning" makes use of a task-dataset-conditioning network to predict the scale and offset parameters of batch normalization layers as in [3, 16, 17]. - Auxiliary tasks are known to be beneficial to few-shot learning [*,**] and learning in general. Moreover, I disagree with the argument (in the conclusion section) that the task sampling technique should be modified for improved performance on a few-shot task, as we should be hesitant about tuning the dataset (which is a feature of the problem) to an algorithm. SPECIFIC COMMENTS pg. 1: "...one aims to...learn a model that extracts information from a set of labeled examples (sample set)..." Introduce the terminology "support set" alongside "sample set". I believe the restriction to the labelled support/unlabelled query setting is not representative of recent works in few-shot learning; e.g., consider [***, ****], which deal with learning with unlabelled data in the support set. pg. 1: It is strange to introduce few-shot learning with Ravi & Larochelle as the first citation, then to claim that the problem has subsequently been reframed by Ravi & Larochelle as meta-learning -- they are the same paper! This needs to be rewritten to correctly capture the nuanced difference between few-shot and meta-learning. pg. 1: The claim that certain approaches to few-shot learning and meta-learning are "influential" and "central to the field" is subjective and pg. 1: "a feature extractor (or more generally a learner)" A feature extractor is more general than a neural network with learned parameters, so this relationship needs to be reversed. Since you consider models with learned parameter, it would be sufficient to rewrite this as "a feature extractor with learned parameters." line 57: "parameterized by \phi, mapping x to z, a representation space of dimension D_z" z is an element of the representation space, not the representation space itself. line 59: "can directly be used to solve the few-shot learning classification problem by association" This needs a more thorough explanation and a more thorough description of the differences between [30] and [25]. In particular, the training criterion for [25] is discussed in lines 62-63 but [30]'s is not. line 60: "proposed to introduce inductive bias" Framing the computation of the class centroid as an "inductive bias" is not useful in this context unless it is identified why it is a useful inductive bias. line 61: "a unique feature representation" The mean of embeddings is not necessarily unique. line 61: "for each class k" It is confusing to use k to index classes when above K have been used to count examples in each class. line 77-8: "This is the case of Matching Networks [30], which use a Recurrent Neural Network (RNN) to accumulate information about a given task." The vanilla version of MNs does NOT use an RNN; only the "full-context embedding" version requires it. line 108-109: "We observed that this improvement could be directly attributed to the interference of the different scaling of the metrics with the softmax." This needs an experimental result, and so likely does not belong in the "Model Description" section. lines 224: "We re-implemented prototypical networks..." Do the experimental results remain the same when employing the authors' original code (https://github.com/jakesnell/prototypical-networks) with the addition of the temperature scaling parameter? [*] Alonso, Héctor Martínez, and Barbara Plank. "When is multitask learning effective? Semantic sequence prediction under varying data conditions." arXiv preprint arXiv:1612.02251 (2016). [**] Rei, Marek. "Semi-supervised multitask learning for sequence labeling." arXiv preprint arXiv:1704.07156 (2017). [***] Finn, Chelsea, Tianhe Yu, Tianhao Zhang, Pieter Abbeel, and Sergey Levine. "One-shot visual imitation learning via meta-learning." In CoRL, 2017. [****] Metz, Luke, Niru Maheswaranathan, Brian Cheung, and Jascha Sohl-Dickstein. "Learning Unsupervised Learning Rules." arXiv preprint arXiv:1804.00222 (2018).

Reviewer 2



This paper introduces a novel prototypical network-inspired architecture for few-shot learning that incorporates several useful modifications that lead to wins in classification accuracy. It analyzes the effect of temperature scaling the output softmax and discusses the nature of gradients in both the low temperature and high temperature regimes. It also proposes a method to condition the embedding network on the few-shot task. Finally, it proposes to train jointly on an auxiliary prediction task. The combination of these modifications leads to significant performance wins on an established few-shot classification benchmark. Overall a strong paper. The contributions of this work are significant. The analysis of scaling presented in section 2.1 is helpful to further understand pathologies that may arise when training prototypical networks and justifies the use of the scaling parameter. Conditioning the embedding network on the task is an intuitive yet unexplored avenue to improved few-shot classification performance. The ablation study in Table 3 shows the contributions of these components as well as the effect of auxiliary training on classification. Moreover, the introduction of the few-shot CIFAR-100 variant will likely prove useful for the community. The paper is very well-written. Experimental procedures are described in sufficient detail and the related work section sufficiently covered relevant material in my opinion. I verified equations 1-4 and they are correct to my knowledge. Were task representations other than the mean prototype experimented with? It seems that higher order statistics of the class representations might be helpful. ===== After Author Feedback ===== I have read the other reviews and the author feedback. Although elements of the modifications explored in this work have previously been proposed in the non few-shot classification setting, I feel there is significance in adapting these ideas to the few-shot scenario and converting them into a win. I therefore maintain my recommendation.

Reviewer 3



# Response to authors’ response # I appreciate the authors’ effort in response. I agree that the derivation and how to implement task conditioning are novel and valuable. However, I do not agree with the authors' claim in the response that they improve a large margin than existing methods when it becomes clear that row 1 of Table 3 is [25] with a stronger architecture. And I do think a detailed discussion on architectures in Table 1 would make the contribution clear. Besides, the auxiliary co-training seems to contribute a lot in Table 3, but the description is short and not clear. The authors must strengthen this part. The task conditioning is not new ([30] has done so) and thus extending the discussion in response is highly recommended. Since the authors replied to most of my comments, I remain the score. My specific response is in [] after each comment. # Summary # This paper investigates how to learn metrics for few-shot learning. The authors theoretically and empirically show the importance of metric scaling in the learning objective. They further show that learning task conditioning metrics could improve the results (by roughly 1 %). The paper provides an insightful introduction and related work to categorize existing few-shot learning algorithms. However, the idea of learning conditioning metrics (or features) seems to be proposed already but the discussion is missing. The experiments also need more explanation to clarify the contributions. # Strengths # S1. The idea of metric scaling is theoretically and empirically justified. Section 2.1 should benefit not only few-shot learning, but the whole metric learning community. S2. The introduction and related work are well organized and inspiring. # Main weakness (comments) # W1. The idea of learning task conditioning metrics (or feature extractors) has been proposed (or applied) in [30], where they learn f(x|S) and g(x|S) where S is the data of the current task. The authors should discuss the relatedness and difference. More informatively, why the proposed method significantly outperforms [30] in Table 1. [Thanks to the authors’ response. Please include (an “extended” version of) the discussion in the final version. I won’t say that [30] is doing postprocessing, however, since by encoding S into LSTM, the feature extractor does change.] W2. The comparison in Table 1 might not be fair, since different models seem to use different feature extractors. The authors should emphasize / indicate this. The authors should either re-implement some representative methods ([25], [30]) using the same architecture (e.g., Res-12) or implement the proposed method in the architecture of [25], [30] for more informative comparison. Is the first row of Table 3 exactly [25] but with the Res-12 architecture? If not, since the proposed model is based on [25], the authors should include it (with the same architecture of the proposed method) in Table 3. [The authors confirmed. From Table 3, it does show that the proposed method still outperforms [25] by 2~3% (compared to the first row). But considering how changing the architecture can largely improve the performance (i.e., comparing Table 1 [25] and Table 3 row 1), the authors need to discuss the architecture of methods in Table 1 in detail. For now, the improvement in Table from [25] to Ours might be misleading.] W3. In Table 3, it seems that having scaling doesn’t significantly improve the performance. Can the authors discuss more on this? Is it because the Euclidean metric is used? If so, the footnote 6 might be misleading because scaling should help a lot for the cosine similarity. [The authors respond. Thanks.] W4. Section 2.4 on the auxiliary task is not clear. Is the auxiliary task a normal multi-way classification task? It would be great to have the detailed algorithm of the training procedure in the supplementary material. [No authors' response.] # Minor weaknesses (comments) # W5. The idea of scaling (or temperature) is mentioned in some other papers like Geoffrey Hinton et al., Distilling the Knowledge in a Neural Network, 2015 It would be great to discuss the relatedness. [The authors responded. Please include it in the final version.] W6. The way FC 100 is created seems to favor learning task conditioning metrics, since now each task is within a superclass, and the difference between tasks (e.g., from different superclasses) will be larger than that if we sample the tasks uniformly from 100 classes. It would be inspiring to compare the improvement of the proposed methods on FC 100 and a version where a task is sampled uniformly. [No authors' response.] W7. Why the experiment on the popular Omniglot dataset is not conducted? [The authors provide explanation. But it would be great to include it in the final version.] W8. Please add the dataset reference to Table 1. Please include background (i.e., algorithm or equation) on [25] in Section 2.1 since it is the baseline model used in the paper.